# Automatic Tree Height Measurement Based on Three-Dimensional Reconstruction Using Smartphone

**DOI:** 10.3390/s23167248

**Published:** 2023-08-18

**Authors:** Yulin Shen, Ruwei Huang, Bei Hua, Yuanguan Pan, Yong Mei, Minghao Dong

**Affiliations:** School of Computer, Electronics and Information, Guangxi University, Nanning 530004, China; 1805170121@gxu.edu.cn (Y.S.); huabei111@163.com (B.H.); 2107310311@gxu.edu.cn (Y.P.); 2101300350@gxu.edu.cn (Y.M.); 2103110245@gxu.edu.cn (M.D.)

**Keywords:** tree height measurement, depth estimation, image segmentation, feature fusion, three-dimensional reconstruction, depth learning, smartphone

## Abstract

Tree height is a crucial structural parameter in forest inventory as it provides a basis for evaluating stock volume and growth status. In recent years, close-range photogrammetry based on smartphone has attracted attention from researchers due to its low cost and non-destructive characteristics. However, such methods have specific requirements for camera angle and distance during shooting, and pre-shooting operations such as camera calibration and placement of calibration boards are necessary, which could be inconvenient to operate in complex natural environments. We propose a tree height measurement method based on three-dimensional (3D) reconstruction. Firstly, an absolute depth map was obtained by combining ARCore and MidasNet. Secondly, Attention-UNet was improved by adding depth maps as network input to obtain tree mask. Thirdly, the color image and depth map were fused to obtain the 3D point cloud of the scene. Then, the tree point cloud was extracted using the tree mask. Finally, the tree height was measured by extracting the axis-aligned bounding box of the tree point cloud. We built the method into an Android app, demonstrating its efficiency and automation. Our approach achieves an average relative error of 3.20% within a shooting distance range of 2–17 m, meeting the accuracy requirements of forest survey.

## 1. Introduction

To establish an efficient forestry resources management system, it is necessary to timely obtain various reliable forest resources data, which puts high requirements on the forest resource survey [1,2]. Tree height measurement is an indispensable aspect of forest resource inventory. Tree height is often used for estimating forest age, volume, biomass and carbon storage [3]. The accuracy of tree height measurements has a significant impact on the validity of subsequent analysis and research [4].

In traditional forest resource surveys, measuring tree height requires manual methods, such as using measuring tapes, which are laborious and time-consuming. These methods may also cause errors due to investigators’ inexperience and potentially harm the trees being measured. High-precision measuring instruments like total stations and theodolites have emerged with the development of forestry information technology [5]. However, these instruments are costly, inconvenient to transport, and have complex operating procedures, making them unsuitable for widespread use in forest surveys.

A promising technique is utilizing remote sensing technologies to estimate tree height through tree reconstruction models, commonly acquired through LiDAR. Mayamanikandan et al. [6] extracted tree volume from terrestrial laser scanning (TLS) data, with an average relative error of 5.31%. Tian et al. [7] registered point clouds obtained from TLS and drone imagery and extracted tree heights in high density coniferous plantations using a canopy height model, with a root mean square error of 6 cm. Yang et al. [8] used 3D laser scanner to obtain 3D point cloud data of trees and automatically calculated tree metrics based on the convex hull algorithm. Nevertheless, it should be noted that the point cloud data of trees are extensive, which renders the subsequent post-processing procedures highly intricate and time-consuming.

Another remote sensing method for tree height measurement involves using image-based 3D reconstruction. Collazos et al. [9] employed a monocular camera and tructure from Motion (SfM) to measure tree height, yielding an average relative error of approximately 7.46%. However, it requires over 20 highly overlapping images and takes an average of 50 min for 3D reconstruction. Lian [10] created a 3D point cloud model of a forest sample plot by extracting overlapped frames from captured videos. This method yielded relative errors ranging from 3.14% to 8.61%. However, it requires calibration targets and involves complicated video shooting and processing. Sun [11] collected images of trees from various angles and used SfM algorithm to transform them into point clouds. This method allowed for extraction of tree structural parameters with high precision. However, it requires complex image collection and long image processing periods.

In recent years, research on machine vision has emerged continually. Machine vision can be mainly categorized into binocular and monocular vision. Zhang et al. [12] proposed a binocular camera-based method for measuring tree height, which is relatively easy to operate with an error rate of no more than 3.93%. Yin et al. [13] captured tree images using binocular cameras, obtained the corresponding 3D point cloud, and finally measured tree structural parameters using the enumeration of the most-valued traversal method with small errors. However, the binocular vision system needs to address the optimal distance between two cameras and feature matching issues, which requires high accuracy and is difficult to achieve.

In contrast to binocular vision, the monocular vision employs a single camera for image acquisition, which is more portable and conducive to data processing. Gao et al. [14] measured tree height based on optical imaging theory, using a single-lens smartphone to capture images. They combined saliency analysis with Canny operator to enhance measurement robustness. However, complex forest environments resulted in significant measurement errors. Wu et al. [15] proposed a monocular vision measurement method based on smartphones, using an improved frequency-tuned saliency algorithm to segment tree images, which has strong robustness but slow computational speed. Coelho et al. [16] used a monocular camera to take two photos of each tree, calculated tree height based on geometrical relationships, and achieved an average relative error of 10.9%. However, this method requires a specially designed calibration object to be placed near the tree for image processing and camera calibration.

Some scholars have applied deep learning to forestry image processing. Juyal et al. [17] used Mask R-CNN to compute tree volume efficiently, but a white rectangular board is required as a reference when photographing trees. Itakura et al. [18] proposed a method combining stereo cameras and YOLO v2 to estimate tree diameter with an RMSE of about 3.14 cm, but the method is unstable for trunk diameters larger than 80 cm. Zhang [19] used binocular vision and YOLO v4 to count trees and measure height, achieving high accuracy only under simple tree backgrounds. However, intelligent forestry measurement still faces challenges, including a lack of relevant datasets for research.

In this context, we have applied smartphones to forestry measurement, achieving a highly accurate and intelligent method for real-time tree height estimation. Our approach eliminates the need for pre-work such as placing calibration objects or designing capture routes. It enables rapid and precise depth estimation, thereby reducing the time and complexity associated with 3D reconstruction and further enhancing the efficiency of tree height measurement. The key contributions of this study are as follows: (1) We captured 300 images of different tree species in various environmental conditions within the campus and annotated the tree regions. Utilizing data augmentation techniques, we created a tree image dataset comprising 1000 pairs of images and made it available for download. This dataset can be used to train other image segmentation model, aiming to alleviate the scarcity of tree image segmentation datasets and further advance the field of tree image segmentation; (2) By combining ARCore and MidasNet, we estimated absolute depth, which is crucial for accurate 3D construction; (3) we improved Attention-UNet for precise tree image segmentation, enabling us to accurately extract tree regions from the images; (4) by integrating color images and depth maps, we generated a 3D point cloud of the scene. Leveraging tree masks and radius filtering techniques, we extracted the 3D point cloud specific to the trees; (5) we measured the tree height by extracting axis-aligned bounding boxes from the 3D point cloud of the trees; (6) we developed an automated tree height measurement prototype app called TreeHeight.

## 2. Materials and Methods

### 2.1. Tree Materials

This study was conducted on trees located on the campus of Guangxi University in China (22°50′ N, 108°17′ E). The campus boasts a variety of tree species, with green spaces in abundance. To evaluate the accuracy of the proposed method, we selected 110 different trees of varying heights and species, as shown in Table 1.

### 2.2. Measurement of True Tree Height

Each tree was measured three times using the ultrasonic rangefinder Vertex IV (Haglöf Sweden AB, Långsele, Sweden) and the transponder T3 (Haglöf Sweden AB, Långsele, Sweden), with the average value taken as the true value. The Vertex IV utilizes ultrasonic waves to measure distances. When measuring tree height, the transponder T3 is first placed at a distance of 1.3 m from the tree. Then, the Vertex IV aligns with transponder T3 to emit ultrasonic waves for calculating the horizontal distance. Subsequently, the highest point of the tree is targeted to emit ultrasonic waves in order to obtain the inclination angle and distance. Finally, the tree height is calculated using trigonometric functions.

### 2.3. Methods

Figure 1 illustrates the proposed method on a sample tree image. First, a color image and absolute depth map are captured using a smartphone with ARCore version 1.36.0 (Google LLC, Mountain View, CA, USA) [20]. Second, MidasNet predicts a relative depth map, which is aligned with absolute depth for a refined absolute depth map. Third, an improved Attention-UNet segments the tree image. Next, a 3D point cloud of the scene is generated by fusing color and depth data. Subsequently, the 3D point cloud of the tree is extracted by applying tree masks to the scene’s point cloud, and noise points are removed by employing radius filtering. Finally, the tree height is computed by extracting the axis-aligned bounding box of the tree’s 3D point cloud.

#### 2.3.1. Depth Estimation

The purpose of depth estimation is to obtain the distance from objects to the camera in images, which is the foundation of 3D reconstruction.

Step 1:Absolute Depth Calculation with ARCore

We use the ARCore SDK to achieve rapid absolute depth estimation on smartphones. ARCore, developed by Google, is an augmented reality framework that equips developers with essential technologies and APIs to create high-quality AR applications for mobile devices. The depth estimation API within ARCore utilizes a single camera and employs the Depth from Motion (DfM) [21]. By analyzing image information, this technique enables the calculation of distances and provides depth values for each pixel. The processing pipeline of DfM is depicted in Figure 2. The algorithm tracks the six degree-of-freedom (6DoF) pose and obtains a grayscale image sequence. Suitable keyframes are selected for stereo matching and rectification using relative 6DoF poses. Then, disparities are estimated using pairwise conditional random fields while eliminating errors with occlusion-aware loss [22]. The resulting sparse depth map is smoothed via planar bilateral filtering for a dense depth map with pixel values corresponding to object distance (in millimeters). The detectable depth ranges from 0 to 65,535 mm, meeting the needs of tree 3D reconstruction.

Step 2:Relative Depth Estimation with MidasNet

Although the DfM algorithm can obtain absolute depth, the output depth map has blurred edges, which is not conducive to reconstructing a 3D model. Therefore, we use the MidasNet to estimate a relative depth map with clearer edges. The generated absolute depth map from the DfM algorithm is then fused with the estimated relative depth map to obtain an absolute depth map with clear edges. The structure of MidasNet is shown in Figure 3, where the encoder extracts image features and the decoder maps the feature maps to the depth map through deconvolution and feature fusion. The encoder part introduces residual blocks from ResNet101 [23].

In order to ensure the accuracy and robustness of the depth estimation model, we combined three datasets for monocular relative depth estimation, namely MegaDepth [24], ReDWeb [25], and WSVD [26], resulting in a total of 3600 pairs of images. As the depth scales differ among these datasets, normalization is required to unify the range of depth values to [0,255]. The dataset was divided into training, validation, and testing sets in a ratio of 8:1:1, with respective sample sizes of 2880, 360, and 360. The loss function is defined as follows:(1)$$L\left(d,d^{*}\right)=\frac{1}{2M}\sum_{i=1}^{M}{\left(d_{i}-d_{i}^{*}\right)}^{2},$$
where *M* represents the number of pixels in the image with valid ground truth depth values, while d and $d^{*}$ denote the predicted and true depth values, respectively.

Table 2 shows the main parameters for training the depth estimation network.

Step 3:Depth Alignment

The depth map obtained by the DfM algorithm can accurately reflect the distance from object surface to camera, but with blurred edges, which are not conducive to 3D reconstruction. On the other hand, the depth map estimated by MidasNet can better preserve edge information, but the scale of depth is unknown, making it impossible to recover the true size of the trees. Therefore, we proposed a method to align the scale of relative depth map using an absolute depth map. There exists a mapping relationship between the relative depth and absolute depth as shown in Equation (2).
(2)A=S⋅R+D,
where A, S, R, and D represent absolute depth, scale factor, relative depth, and displacement coefficient, respectively. Firstly, the two depth maps obtained from Steps 1 and 2 are converted into one-dimensional representations. This is achieved by traversing the pixels in the maps sequentially from left to right and top to bottom. Subsequently, the least squares method is employed for data fitting to estimate the values of S and D. By substituting the values of S, R, and D into Equation (3), the edge-refined absolute depth map can be obtained.

#### 2.3.2. Tree Image Segmentation

By acquiring a tree mask through image segmentation, we are able to extract corresponding point clouds of the trees from the three-dimensional point cloud of the scene for measuring tree height.

Tree Image Segmentation Dataset Creation

We annotated 300 tree images for training and testing the image segmentation network. Each training data set contains three images: a color image, a relative depth map, and a tree mask. By using data augmentation techniques such as flip transformation, random cropping, and scale transformation, we enlarged the dataset to 1000 images and divided them into training set, validation set, and test set at a ratio of 8:1:1.

2.Image segmentation model selection

Currently, there are several image segmentation models available, including Mask R-CNN [27], HRNet [28], DeepLabv3+ [29], UNet [30], and Attention-UNet [31]. To compare their performances, we trained all five models on the same tree image dataset with identical hyperparameters (Table 3), and tested them on the same test set. Some of the segmentation results are illustrated in Figure 4.

In scenario (a), all five networks were able to accurately segment the tree crowns due to their distinguishable colors from the background. However, the trunks share similar colors with the background, and only Attention-UNet was capable of producing satisfactory segmentation results. This is crucial for subsequent point cloud segmentation.

In scenario (b), the ideal segmentation should distinguish the middle and right trees while leaving the crown of the left tree unsegmented. However, HRNet, DeepLabv3+, and UNet failed to segment the right tree and misclassified the crown of the left tree as part of the middle tree. Although Mask R-CNN and Attention-UNet segmented the right tree correctly, they still made errors in crown segmentation, with Attention-UNet having fewer misclassifications.

In scenario (c), due to the interference from the background, all five models exhibited crown mis-segmentation, but Attention-UNet outperformed the other models. Furthermore, due to the white paint on the bottom part of the tree, Mask R-CNN, HRNet, DeepLabv3+, and UNet were unable to segment the full trunk, while Attention-UNet accurately segmented the trunk section.

Overall, Attention-UNet achieves the best segmentation performance among the three cases. By introducing attention mechanisms to highlight local features, this model alleviates the impact of complex backgrounds on tree segmentation to some extent. However, when the tree crowns are intertwined or overlap with each other, the model usually fails to correctly segment them, which impairs the accuracy of tree height measurement. Thus, we adopted Attention-UNet for tree image segmentation and introduced the following modifications:(1)Depth map is introduced as an additional input to Attention-UNet.(2)At each layer, the encoder performs two convolution and activation operations to obtain RGB feature maps and depth feature maps separately, followed by a feature fusion. The fused results are then transmitted to the attention gates of the decoder through skip connections at each stage, enabling complementary utilization of the two modalities.
3.Network Architecture of the Improved Attention-UNet

In this study, we introduce depth maps as an additional input to Attention-Unet and perform multimodal feature fusion. The depth maps exploit the depth differences between trees to provide richer information for the network, alleviating the interference of overlapping crowns on tree image segmentation. The proposed model, called Depth-Attention-Unet, is shown in Figure 5.

4.Training the Improved Attention-UNet

The main parameters used for training the image segmentation network are shown in Table 4.

#### 2.3.3. Tree 3D Reconstruction

In order to measure tree height, it is necessary to recover the 3D structure of the tree from the 2D information. In this paper, we obtained a scene’s 3D point cloud by fusing color images, depth maps, and camera intrinsic parameters. Then, using the image segmentation results, we filtered out the background from the 3D point cloud, and further denoised it to obtain the final tree point cloud for tree height measurement.

Scene 3D Reconstruction

The depth value d estimated in Section 2.3.1 has a unit of millimeters, while the reconstructed 3D point cloud has a unit of meters, requiring a scaling factor, denoted as scale, for unit conversion. In this case, we set scale to 1000. The camera intrinsic parameters, including the camera focal lengths ($f_{x}$, $f_{y}$) and the optical centers ($c_{x}$, $c_{y}$), are obtained when calling the phone camera. According to pinhole imaging principle, the relationship between the 3D point cloud coordinates (x, y, z) and the pixel coordinates (u, v) is shown in Equations (3)–(5):(3)A=scale⋅R+swift
(4)$$y=-\frac{\left(v-c_{y}\right)\cdot z}{f_{y}}$$
(5)$$z=-\frac{d}{scale}$$

2.Point Cloud Segmentation and Denoising

In this research, tree point clouds were extracted using the tree masks obtained through image segmentation. Since each point in the point cloud corresponds to a pixel in the colored image, point cloud segmentation could be easily performed. However, image segmentation can contain certain errors that introduce noise into 3D reconstruction results, thereby interfering with tree height measurements. To mitigate this issue, we adopted a radius filtering approach. For each point in the point cloud, we calculated the number of points within a specified radius. Points with fewer than the specified count were then removed, resulting in the final 3D point cloud of the tree. The values for the radius range and count should be adjusted according to specific conditions. In our experiments, we set the radius range to 1.5 m and the count to 10.

#### 2.3.4. Tree Height Measurement

Axis-aligned bounding boxes were extracted from the point cloud of tree as shown in Figure 6. The length of the box in the Y direction represents the tree height in the real world, with units in meters (m).

#### 2.3.5. App Design and User Experience

In this work, we developed TreeHeight, an Android smartphone-based system for measuring tree height, consisting of a frontend and a backend. The frontend is an Android application developed in Java, incorporating ARCore version 1.36.0, OkHttp3 version 4.4.1 (Square, Inc., San Francisco, CA, USA) [32], Gson version 2.9.1 (Google LLC, Mountain View, CA, USA) [33], and Obj version 0.2.1 [34] to enable data collection, storage, and transmission. The backend, implemented in Python, is built upon Flask version 2.2.5 [35], PyTorch version 1.11.0+cu115 (Linux Foundation, San Francisco, CA, USA) [36], Open3D version 0.17.0 (Linux Foundation, San Francisco, CA, USA) [37], and OpenCV version 4.8.0 (Intel Corporation, Santa Clara, CA, USA) [38], focusing on tree height measurement task.

During the capture process, users only need to slightly move their mobile phones from a single position. This position should be where the tree is visible. By doing so, they enable ARCore to gather more information from a continuous sequence of camera frames, thereby improving the accuracy of depth estimation. User can capture both a color image and depth map of the target tree at a specific moment without the need to photograph the tree from different angles, and then upload the data to the backend for measurement. The app displays the captured images, measured tree height, aligned depth map, tree masks, and 3D point cloud to the user.

The main app screen includes the photo shooting screen and the measurement result display screen, as shown in Figure 7 and Figure 8. When entering the photo shooting screen, the system calls the camera to continuously capture camera frames and estimate absolute depth. The camera intrinsic parameters are obtained and saved by the system when the screen is first opened. User can cancel the shooting at any time by clicking on the arrow in the upper left corner of the screen. When the screen displays the color image as shown in Figure 7a, click the “Show Depth” button, and the system will render the depth map on the screen in real time as shown in Figure 7b. Click the “Hide Depth” button, and the screen will return to displaying the color image.

The user can determine the appropriate time by observing the depth map, save the current camera frame and its corresponding depth map by clicking the “Save” button, and automatically jump to the data details interface, and after 3–4 s, the calculation results are displayed, as shown in Figure 8a. The image segmentation result shows different tree masks filled with various colors, along with the symbols denoting different trees and their corresponding tree heights displayed in the upper-left corner of the masks. The tree heights for each tree are also explicitly written below the image. The user can swipe the screen to view the aligned depth map and 3D point cloud, as illustrated in Figure 8b,c.

## 3. Results

### 3.1. Results of Depth Estimation

ARCore can estimate depth values from 0–65 m, with optimal accuracy at 0.5–15 m. Measurements are reliable up to 25 m, but error increases quadratically with distance [39]. However, weighted averaging during dense reconstruction of sparse depth maps leads to blurred object edges, as shown in Figure 9. Directly using these rough depth maps for 3D model reconstruction results in inaccuracies in reflecting real-world tree scenes and affects the final precision of tree height measurements.

After training, MidasNet achieved an accuracy with threshold of 91.21% when the threshold value was set to 1.25 on the test dataset. Partial relative depth estimation results are presented in Figure 10. Compared with the absolute depth maps obtained through ARCore, the edges of the relative depth maps were observed to be clearer.

The variation curves of depth for the same column of pixels in the aligned depth map obtained by scaling and shifting relative depth and the absolute depth map obtained by DfM are shown in Figure 11. It can be observed that the aligned depth map has the same scale as the absolute depth map.

Figure 12 compares three depth map types. Aligned fine-grained absolute depth maps have the same pixel scale as coarse absolute depth maps from the DfM algorithm, and resemble the fine-grained relative depth map from MidasNet. This indicates the aligned depth map has both the depth scale determinacy of absolute depth maps and object edge clarity of relative depth maps, enabling excellent 3D reconstruction results when fused with color images.

### 3.2. The Training Results of the Tree Image Segmentation Network

To verify the effectiveness of our proposed improvement method for Attention-UNet, we trained Attention-UNet and Depth-Attention-UNet on the same dataset under the same training environment. The performance of the trained models in tree image segmentation tasks was compared on the test set, as shown in Table 5. Partial tree image segmentation results are displayed in Figure 13. Without depth information, Attention-UNet tends to misclassify dark objects overlapped with trees, such as lamp posts, as tree trunks. Also, it fails to accurately identify the edges of tree crowns when they have similar color as the background, and there are boundary adhesion issues with tree crowns. By contrast, Depth-Attention-UNet excludes non-tree areas and differentiates between different trees based on depth features, resulting in more accurate image segmentation.

### 3.3. Results of Tree 3D Reconstruction

The 3D coordinates of each pixel point were calculated using the depth map and camera parameters, and combined with the color information of the corresponding pixels in the RGB image to generate a 3D point cloud, as shown in Figure 14.

Next, the tree mask was used to segment the point cloud as shown in Figure 14, resulting in 3D point cloud of individual trees as shown in Figure 15a. Finally, the radius filter was utilized to remove noisy points from the point cloud, and the denoising effect is shown in Figure 15b.

### 3.4. Accuracy of Tree Height Measurement

Some of the measurement results are shown in Table 6. The relationship between the relative error and the shooting distance is presented in Figure 16. Within the shooting distance range of 2–17 m, the measurable tree height ranges 3.7–24.4 m. The maximum relative error was 4.87%, the minimum relative error was 1.92%, and the average relative error was 3.20%, satisfying the accuracy requirements of the forest resource survey.

### 3.5. Time of Tree Height Measurement

To evaluate the time required for completing a measurement using our developed app, a random tree was selected and measured 10 times using the app. The running time for each step was recorded and tabulated in Table 7.

The average total time for a tree height measurement completed by the backend of the system is 3268.81 ms, or 3.27 s. The data transmission time is within 1 s due to the data size being no more than 1 M. Thus, the time required to obtain measurement results from data upload is under 5 s, indicating that the proposed tree height measurement method is highly efficient and can satisfy practical requirements.

## 4. Discussion

### 4.1. Tree Height Measurement

The experimental results indicate that the 3D reconstruction-based method for measuring tree height is consistent with the traditional ultrasound ranging method. However, the measurement error increases as the shooting distance increases due to the error of depth estimation and edge loss of small objects during the conversion of color and depth images to 3D point clouds. Moreover, some branches and small trunks cannot be accurately reconstructed when the shooting distance is too far. Increasing image resolution can improve measurement accuracy, but it may also increase processing time. Therefore, a suitable resolution and shooting distance should be chosen based on practical considerations.

It is important to note that the proposed methodology and its results are currently in the experimental prototype stage and are not yet suitable for practical application in forestry operations. The tree images used in this study were collected within a university campus, which does not fully represent the complexities of real-world forest environments. As a result, the accuracy of the method may vary significantly depending on factors such as tree types, forest conditions, and geographic locations. At this stage, there hasn’t been sufficient validation across diverse scenarios, especially those representing authentic wild environments. In future research, we plan to conduct thorough validation of the method, including comprehensive testing in various scenarios.

This study employs widely accessible smartphone as a measurement tool, offering greater convenience compared to laser distance meter and binocular camera [9,12,13]. The proposed method for measuring tree height in this study eliminates the need for manual operations, as it only requires taking images. In contrast, optical imaging-based methods [5,9,10,12,13] rely on manually obtaining horizontal distances and placing reference objects, highlighting the higher level of automation achieved in our research. Additionally, our study requires capturing images of trees from a single perspective, resulting in smaller data size and higher processing efficiency compared to capturing multiple angle images or continuous videos [9,10]. Although our study exhibits an average relative error of 3.2% in tree height measurements, slightly higher than the methods using rangefinders [5] or stereo cameras [12,13], it is comparable to or lower than other techniques that only employ monocular cameras as data acquisition devices [10,11,14,16]. Table A1 in Section A.1 presents a comprehensive comparison with previous research.

### 4.2. Depth Estimation

In depth estimation, accurate absolute depth estimation can only be achieved within a limited distance range using the camera and inertial sensors equipped on ordinary smartphones. To address this problem, machine learning algorithms could be employed to train on data collected from the camera and inertial sensors, resulting in more accurate depth estimation models. When measuring tall trees, multi-view image processing methods or image stitching could be utilized to improve the accuracy of depth estimation.

### 4.3. Tree Image Segmentation

We propose an improved method to address inaccurate tree crown boundary detection in tree image segmentation tasks using Attention-UNet. The method introduces depth features and performs multimodal fusion with color features to alleviate the interference caused by complex backgrounds and overlapping tree crowns. The approach effectively copes with these challenges by leveraging the spatial information of the depth features and the complementary information of color and depth maps. Experimental results on a tree dataset demonstrate the effectiveness of this approach, while the results on the PASCAL VOC2007 dataset [40] (Section A.2) confirm its ability to enhance the image segmentation performance of Attention-UNet under dense object distribution. Future work can explore more advanced deep learning network structures to further improve the accuracy of tree image segmentation.

## 5. Conclusions

In order to fulfill the need for efficient, cost-effective, and time-saving tree height acquisition in forest survey, we proposed an automated tree height measurement method based on 3D reconstruction and conducted experiments using various tree species to evaluate the effectiveness of the method. Firstly, the absolute depth map was obtained by combining ARCore and MidasNet. Then, an improved Attention-UNet was employed for tree image segmentation. The obtained depth map was fused with the color image to generate a 3D point cloud. Subsequently, the tree mask was utilized to extract the 3D point cloud corresponding to the trees. The radius filter was applied to the point cloud in order to eliminate noise points. Finally, the axis-aligned bounding box of the point cloud is extracted, and the height of the bounding box in the *Y*-axis direction of the world coordinate system is taken as the tree height. This measurement was then compared with values obtained using Vertex IV. The results show that the average relative error of tree height measurement is 3.20%, which meets the requirements of forest precision surveys.

## Figures and Tables

**Figure 1 sensors-23-07248-f001:**
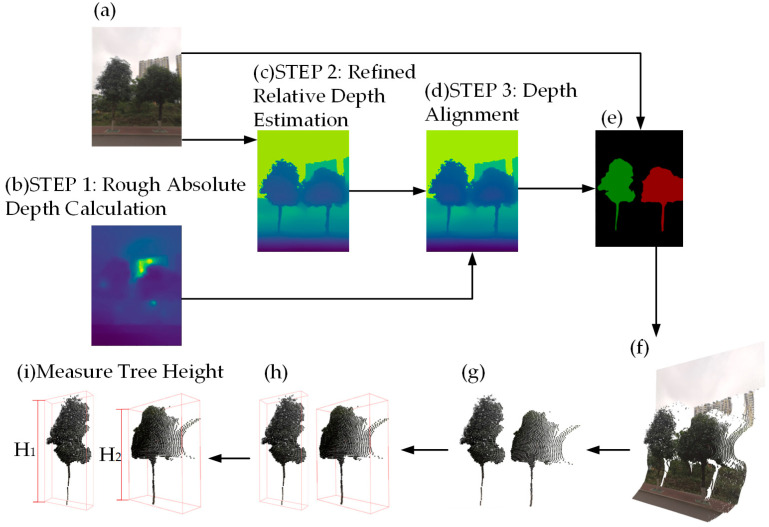
Tree height measurement process. The RGB image is shown in (**a**). The sub-steps of acquiring fine absolute depth map are displayed in (**b**–**d**). The tree mask is shown in (**e**). The 3D point cloud of the scene is shown in (**f**). The segmented 3D point cloud of the tree is shown in (**g**). The extracted axis-aligned bounding box of the 3D point cloud of the tree is shown in (**h**). The final obtained tree heights, H1 and H2, are shown in (**i**).

**Figure 2 sensors-23-07248-f002:**
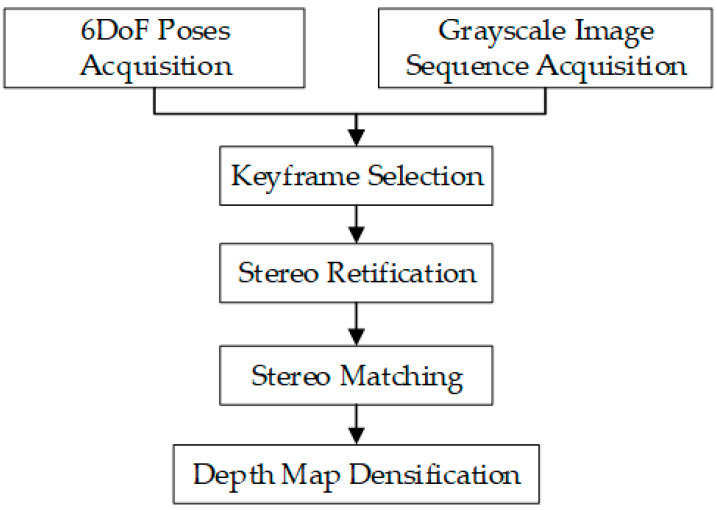
DfM algorithm process.

**Figure 3 sensors-23-07248-f003:**
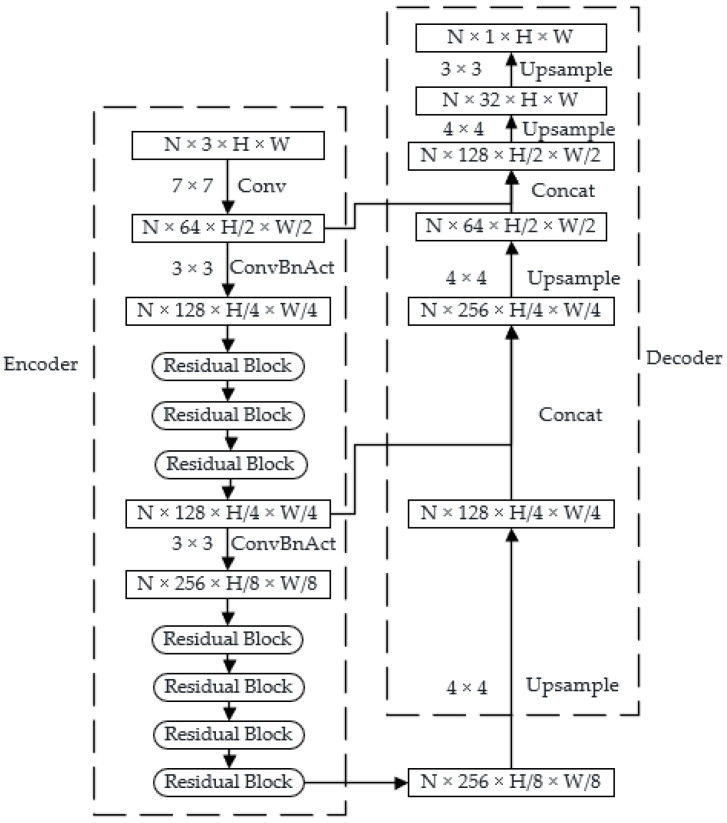
Structure of MidasNet.

**Figure 4 sensors-23-07248-f004:**
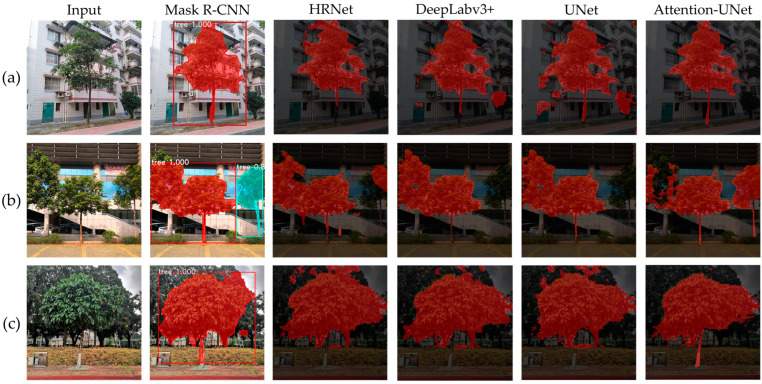
Comparison of the segmentation performances of the five networks in different scenarios. (**a**) The background is simple and only one tree exists. (**b**) The crowns of the left and middle trees overlapped with similar colors, the trunk of the left tree was not visible, and the trunk of the right tree blended with the background. (**c**) The tree crowns have similar colors to the background, and the trunks are partially painted white.

**Figure 5 sensors-23-07248-f005:**
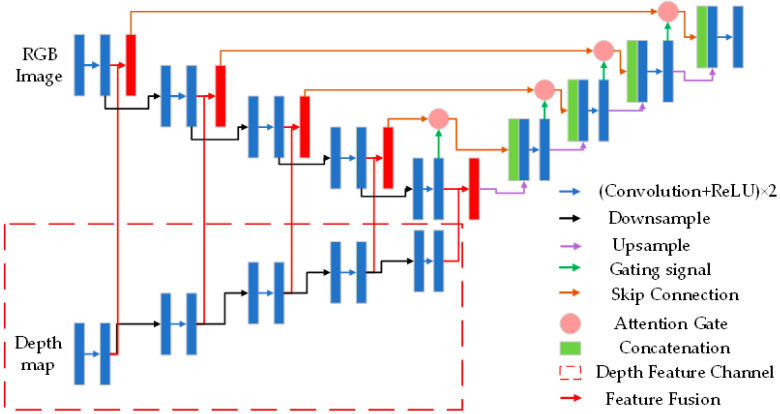
Structure of Depth-Attention-Unet. The section enclosed by the dashed box represents the newly added depth feature channel.

**Figure 6 sensors-23-07248-f006:**
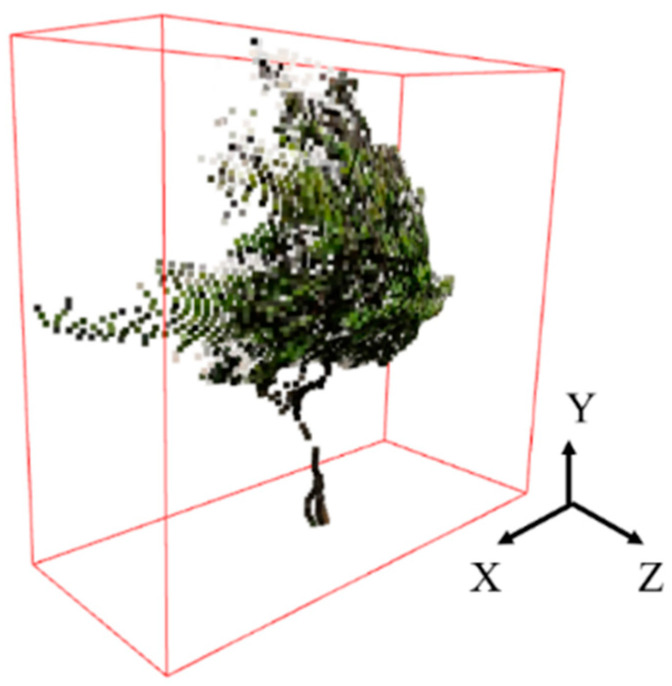
Axis alignment bounding box of tree point cloud.

**Figure 7 sensors-23-07248-f007:**
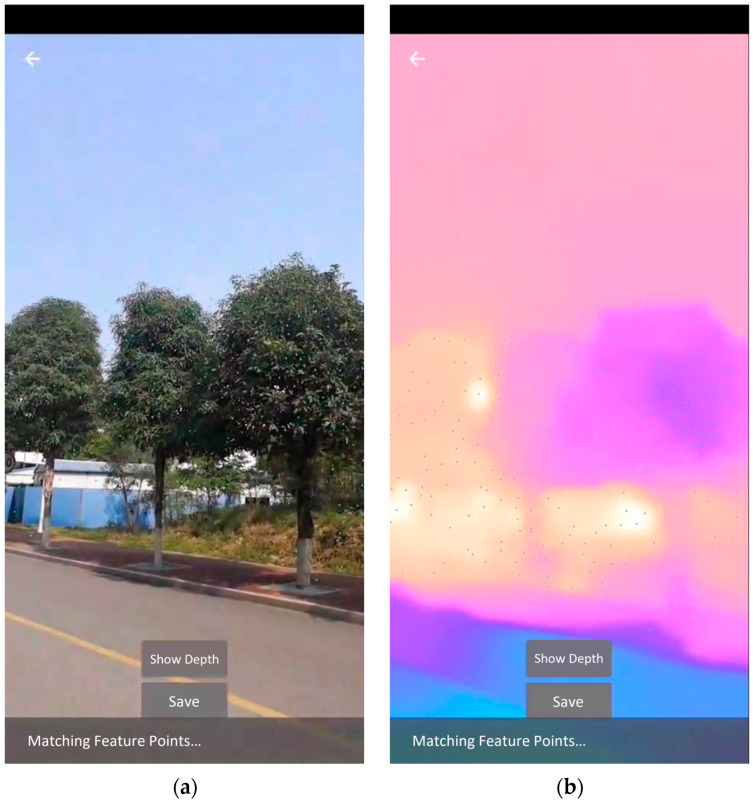
The photo shooting screen. (**a**) The app shows the color image; (**b**) After clicking ‘Show Depth’, the app displays the depth map.

**Figure 8 sensors-23-07248-f008:**
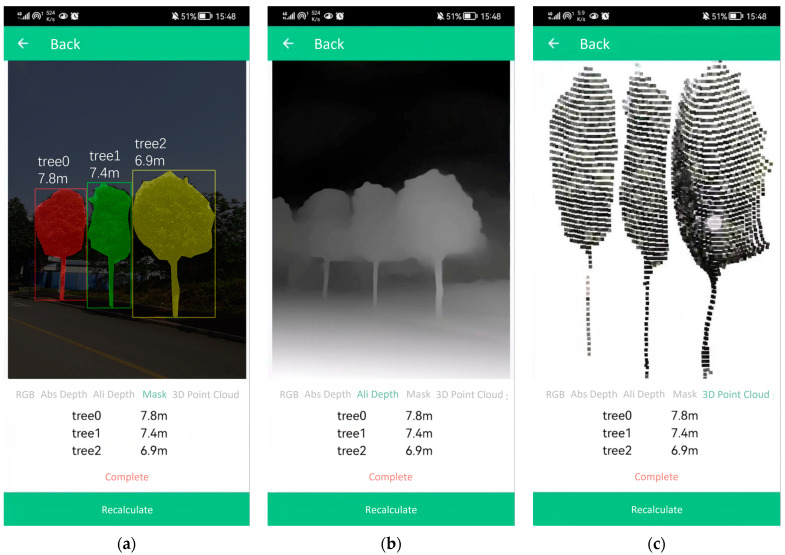
The measurement result display screen. (**a**) The tree image segmentation result; (**b**) The aligned absolute depth map; (**c**) The 3D point cloud of the trees.

**Figure 9 sensors-23-07248-f009:**
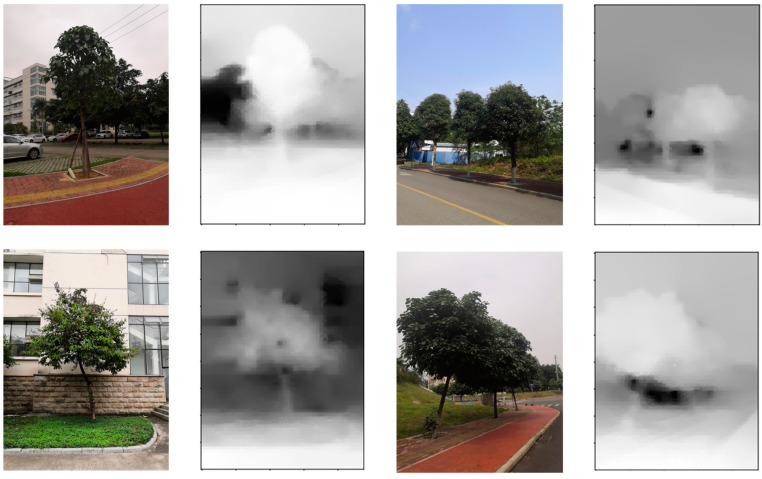
Rough absolute depth map.

**Figure 10 sensors-23-07248-f010:**
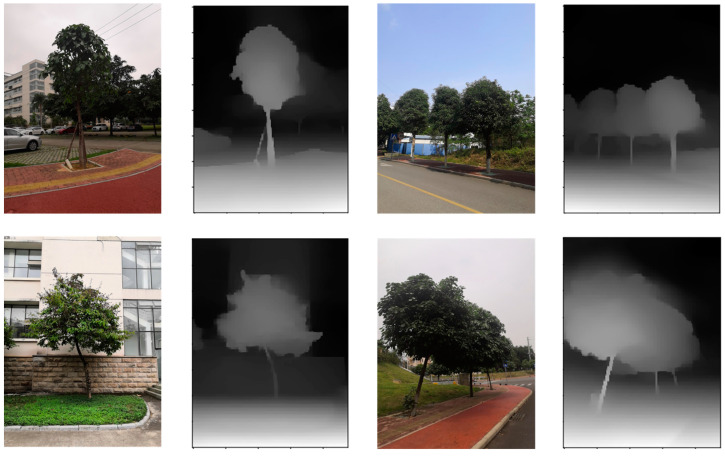
Fine relative depth map.

**Figure 11 sensors-23-07248-f011:**
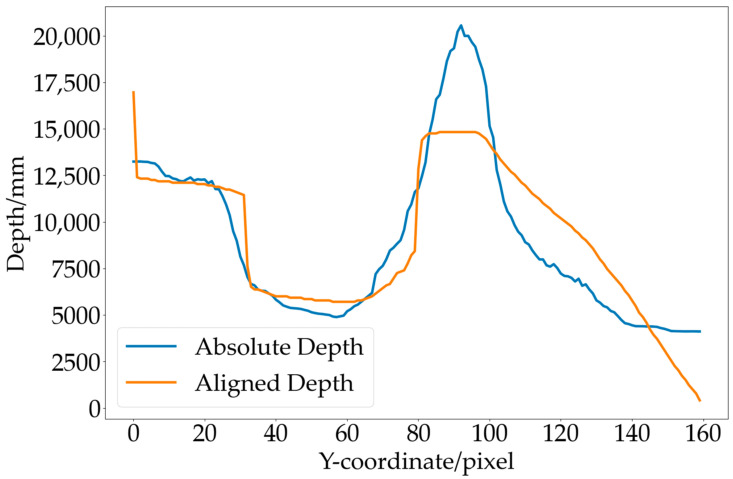
Depth alignment results.

**Figure 12 sensors-23-07248-f012:**
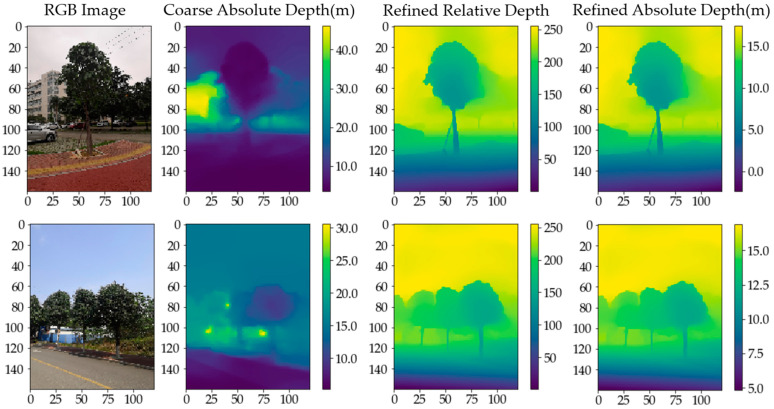
Comparison of depth maps.

**Figure 13 sensors-23-07248-f013:**
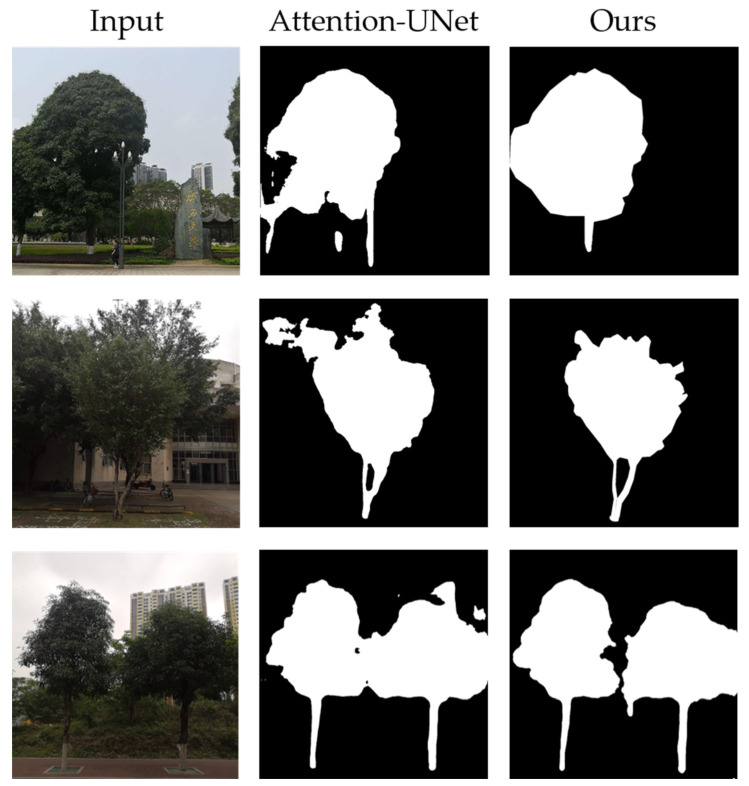
Comparison of Tree Image Segmentation Performance.

**Figure 14 sensors-23-07248-f014:**
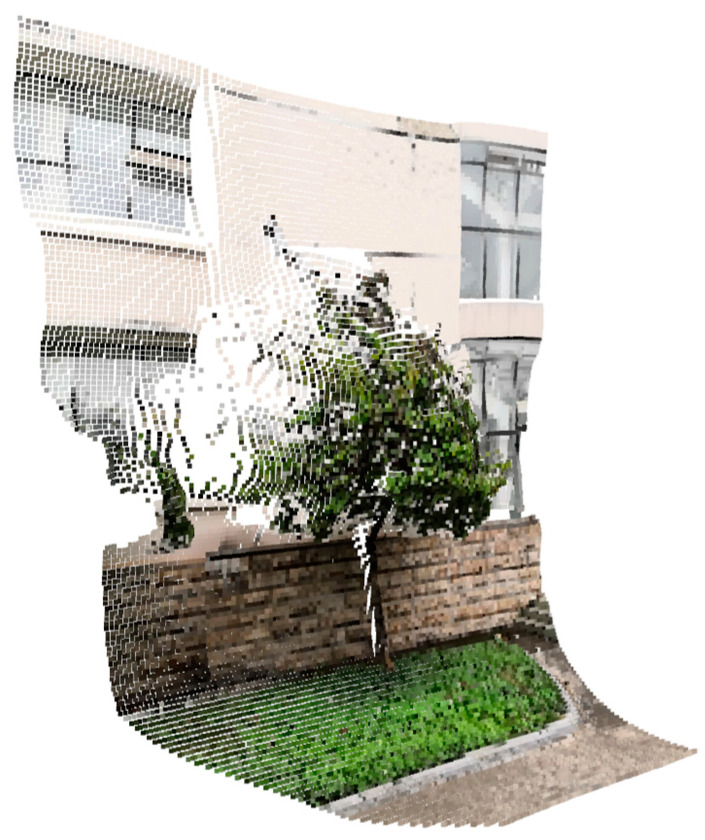
3D point cloud of the scene.

**Figure 15 sensors-23-07248-f015:**
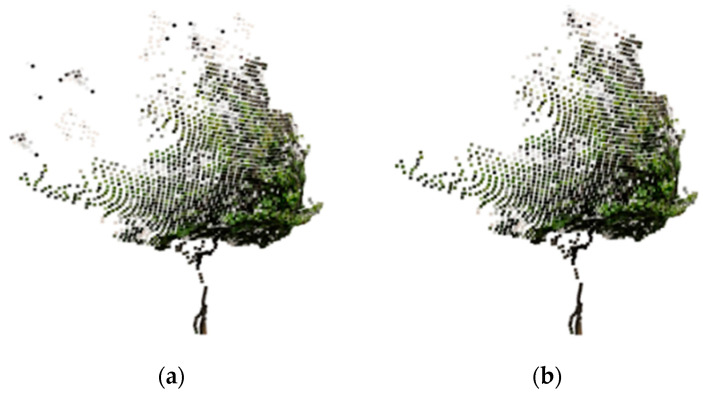
Comparison before and after point cloud denoising. (**a**) The point cloud before denoising; (**b**) The point cloud after denoising.

**Figure 16 sensors-23-07248-f016:**
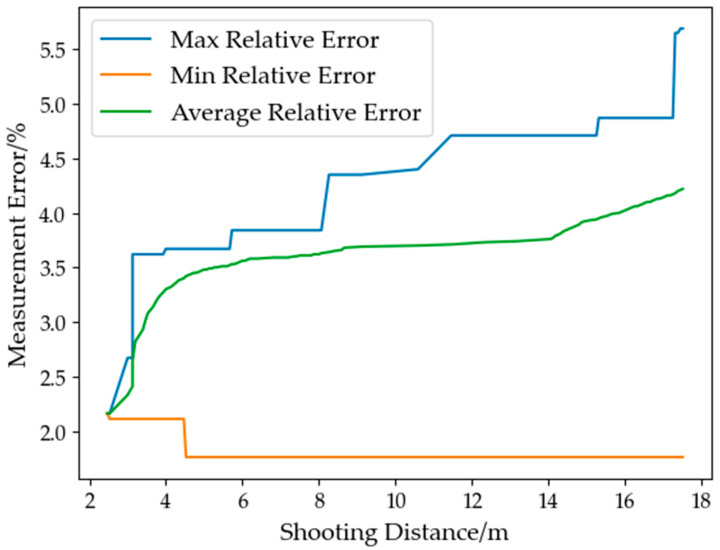
The trend of measurement error variation.

**Table 1 sensors-23-07248-t001:** Quantitative Information regarding selected trees.

Species	Number of Trees	Min/Max Tree Height (m)
*Sterculia nobilis*	26	3.7/5.1
*Taxus chinensis*	19	25.1/26.0
*Alstonia scholaris*	17	17.2/21.3
*Mangifera indica*	14	9.6/13.7
*Sindora tonkinensis*	12	19.8/24.4
*Bombax malabaricum*	11	21.1/23.8
*Trachycarpus fortunei*	6	5.1/18.5
*Areca catechu* L.	5	13.7/24.1

**Table 2 sensors-23-07248-t002:** Setting of training parameters for depth estimation model.

Parameter	Value
Input size	384 × 384
Batch size	8
Initial learning rate	0.0001
Total iteration steps	300

**Table 3 sensors-23-07248-t003:** Setting of hyperparameters for image segmentation models.

Parameter	Value
Input size	512 × 512
Batch size	8
Initial learning rate	0.001
Total iteration steps	100

**Table 4 sensors-23-07248-t004:** Setting of training parameters for image segmentation model.

Parameter	Value
Input size	512 × 512
Batch size	8
Initial learning rate	0.001
Total iteration steps	200

**Table 5 sensors-23-07248-t005:** Comparison of tree image segmentation performance.

Model	IoU (%)	PA (%)
Attention-UNet	91.20	96.27
Ours	95.31	98.14

**Table 6 sensors-23-07248-t006:** Tree height measurement data.

Sample No.	True Value/m	Measured Value/m	Shooting Distance/m	Relative Error/%
1	3.7	3.78	2.47	2.16%
2	4.8	4.63	3.20	3.54%
3	5.6	5.42	3.73	3.21%
4	6.5	6.73	4.33	3.54%
5	7.4	7.63	4.93	3.11%
6	8.7	9.02	5.80	3.68%
7	9.3	9.58	6.20	3.01%
8	10.8	11.04	7.20	2.22%
9	11.3	11.63	7.53	2.92%
10	12	12.23	8.00	1.92%
11	13.7	14.12	9.13	3.07%
12	15.9	15.2	10.60	4.40%
13	17.2	16.39	11.47	4.71%
14	18.5	19.32	12.33	4.43%
15	19.8	20.66	13.20	4.34%

**Table 7 sensors-23-07248-t007:** Time of tree height measurement.

Sample No.	Depth Estimation/ms	Tree Image Segmentation/ms	3D Reconstruction/ms	3D Point Cloud Segmentation/ms	Denoising/ms	Tree Height Extraction/ms	Overall Process/ms
1	1483.10	1809.27	6.09	3.00	11.27	0.09	3312.82
2	1531.66	1981.08	6.42	3.00	10.63	0.10	3532.89
3	1422.50	1845.22	6.88	3.00	13.28	0.10	3290.98
4	1455.29	1709.31	6.29	3.00	14.83	0.10	3188.81
5	1512.97	1581.32	6.38	3.00	11.20	0.10	3114.97
6	1467.25	1627.70	6.46	3.00	13.01	0.10	3117.52
7	1426.75	1936.47	6.69	2.99	10.81	0.10	3383.81
8	1436.96	1897.49	6.24	2.99	10.77	0.10	3354.54
9	1520.31	1503.55	6.18	2.99	11.14	0.10	3044.27
10	1466.61	1859.39	6.83	3.00	11.55	0.09	3347.47
Average	1472.34	1775.08	6.45	3.00	11.85	0.10	3268.81

## Data Availability

The source code of the prototype app is publicly available on GitHub at https://github.com/LisaShen0509/Tree_Height_Measurement (accessed on 27 July 2023). Tree image dataset is available at https://drive.google.com/file/d/1kG6LWMOAiA2KvGF_suZ5cG_4C-udUV0m/view?usp=sharing (accessed on 27 July 2023).

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
