# Peer review of "Automatic Tree Height Measurement Based on Three-Dimensional Reconstruction Using Smartphone"

_sensors, 2023, doi:10.3390/s23167248_

Round 1
Reviewer 1 Report
The authors of this manuscript propose a method for tree height estimation in forest surveys. The proposed algorithm is carried out in the following stages:
1. An absolute depth map is generated using ARCore and MidasNet
2. A variation of the Attention-UNet is employed to segment the image of the tree
3. The color image is fused with the depth map, generating a 3D point cloud
4. A fitting curve is computed to estimate the highest point on an occluded tree in the image. This allows for the successful extraction of the 3D point cloud of the tree in the image.
5. A radius filter is used to remove noise pixels
6. A bounding box is created, where its X and Y axes are aligned and its height (highest Y point) is assumed to be the tree height.
Overall, this manuscript is relatively well written. The references used are adequate and recent. The introduction section is clear, paving the way to the reader for good comprehension of the context and the state-of-the-art. The same could be said about the material and methods section. The results section provides both a qualitative and a quantitative analysis of the experimentation. However, not enough comparison with the literature is provided here. The discussion section provides appropriate commentary on the computed results. Finally, the conclusions section summarizes the main ideas of the manuscript, the proposed algorithm and the main result, confirming the suitability of the proposed algorithm to actual usage in forests’ surveying.
My comments are related to the rather limited comparisons in the results section, as well as to the use of the English language in the manuscript. Clearly, a language proof-reading is needed. A few examples of the mistakes found include:
· Unneeded use of hyphens in some words, for example in lines 54 and 57.
· Incomplete sentences on lines 73 and 74.
· Spelling mistake in line 81.
· Grammar mistake in lines 146 and 147.
Other than that, I congratulate the authors on their work!
A few examples of the mistakes found include:
· Unneeded use of hyphens in some words, for example in lines 54 and 57.
· Incomplete sentences on lines 73 and 74.
· Spelling mistake in line 81.
· Grammar mistake in lines 146 and 147.
Author Response
Response to Reviewer 1 Comments
Point 1: Limited comparisons in the results section.
Response 1: Thank you very much for your suggestions. In Section 4.1, we have added a comparison with previous relevant studies, which is organized into Table A1.
Point 2: The use of the English language in the manuscript.
Response 2: Thank you very much for your correction. We have reviewed and rectified the unnecessary hyphens, incomplete sentences, as well as spelling and grammar mistakes.
Reviewer 2 Report
Section 2.1. It would be great if you could add the geographical location coordinates in the text or through the Figure.
Line 177. It should be Figure 5.
Section 2.2.5. The section lacks information about the name of the app and its availability (link to download, free of charge or paid, etc.). Besides, it would be beneficial if you briefly described the process of app building (what engine or kit was used, what language of programming, etc.). And the last question: whether an iOS app is going to be created? Or this is an exclusively Android-based product.
Reviewer 3 Report
Review of the paper "Automatic Tree Height Measurement Based on Three-dimen-2 sional Reconstruction Using Smartphone"
Summary of the paper: This paper proposes a new method for automatically measuring tree height using three-dimensional (3D) reconstruction, with the aim to facilitate efficient forestry resource management. The authors' method utilize a smartphone to capture color images and depth maps of trees. These are then used in a 3D reconstruction process to estimate tree height. The technique involves combining ARCore and MidasNet network to estimate absolute depth and an improved Attention-UNet network for image segmentation of the tree. The method has been built into an Android app and achieves an average relative error of 3.20% within a shooting distance range of 2-17m.
Comments:
The methodology proposed for depth estimation, which involves deriving both absolute and relative depth, followed by the application of an axis-aligned bounding box to the point cloud, is a commendable conceptual framework. The integration of depth maps with a modified Attention-UNet for performing multimodal feature fusion is innovative and worth noting.
The paper is written with clarity and employs appropriate technical language.
However, two primary concerns are evident:
1. The technique of 3D reconstruction of partially occluded trees, based on trunk direction extrapolation and curve fitting, is highly susceptible to inaccuracies. This methodology might function appropriately under certain ideal conditions but lacks robustness under complex scenarios. Attempting to estimate the height of a tree with significant occlusions could lead to substantial errors during practical implementation. Issues such as understory shrubs and herbs can obstruct the visibility of trunks, further compromising the reliability of curve fitting. The potential errors for partially occluded trees could be significantly high. After a careful assessment of the advantages and disadvantages of this approach, it is suggested that this concept be completely removed from the paper due to its instability.
2. It should be explicitly stated in the paper that the proposed methodology and the corresponding results are in the experimental prototype stage and are not yet ready for practical deployment in forestry operations. There is a conspicuous absence of comprehensive validation. The accuracy of the methodology could vary substantially based on the type of trees, forest conditions, and geographic locations. The study, which is based on images captured within a university campus, essentially represents a controlled laboratory environment rather than the complex, real-world forest environments. Hence, comprehensive validation across a diverse range of scenarios, particularly those representing authentic wild environments, has not been adequately carried out.
Reviewer 4 Report
Overall, this study was nicely organized and the paper was very well written. The abstract is excellent and you have done a particularly good job of establishing motivation for your work by identifying the limitations of existing techniques. The novelty and algorithm development are also nicely documented. I just have some minor concerns that should be addressed prior to publication:
-Construction of the depth maps is a little unclear in the first half of the paper. It almost seems as though you are using LiDAR, since you refer often to point clouds. I suggest explaining the function of ARCore and MidasNet earlier in the text.
-You mention occluded trees in the introduction, and the description seems to suggest the trees are being obscured by other objects. Perhaps you should specify that the leaves and branches for a given tree are obscuring its highest point.
-You mention this technique works for camera distances of 2-17 m? What is the corresponding tree height range that is possible with this scale? It seems a tree would need to be relatively short (perhaps < 3 m) to image from a distance of 2 m. Is the user required to scan the phone through various angles to capture the full height?
-In Section 2.2.2, you describe extracting tree point clouds from scene point clouds. However, the text seems to suggest this step was performed after implementation of the attention U-Net. However, earlier in the paper, you suggested these point cloud data were provided as input to the U-Net, which seems like a contradiction. In other words, the point cloud could not have been extracted by a network that required the point cloud as input. This should be clarified.
-There is no mention of a ground truth until the discussion section. Your approach to measuring the true heights of trees needs to be referenced earlier in the paper, perhaps when describing network training.
-The source of the training data is unclear for the first half of the paper. I suggest mentioning in the introduction section that you collected photographs of trees as part of the study.
-I found a few typographical errors in the text. For example, lines 44, 49, 52, 57, 74, 147, and 402.
-Your literature review does a fantastic job of identifying the limitations of other studies. However, the novelty of your work is not stated explicitly. I suggest adding a sentence or two, emphasizing the uniqueness of your approach. Which benefits does it provide?
-On line 88, you mention a lack of datasets for use in this field. Is this one of the problems you are trying to address with this study? It is never mentioned as one of the goals or objectives for your work.
-Section 2.1 lists several tree species, but no quantitative information is included. What was the range of tree heights? Do you know how many samples you collected from each species? Perhaps you could add a table or a histogram?
-You refer to noise filtering on multiple occasions, but little detail is provided. Line 527 mentions a radius filter, but I don’t believe this was discussed anywhere else in the text. How was this filtering performed?
-Line 222 mentions the collection of 100 images. However, you also refer to the use of 3600 samples and (separately) the use of 3000 samples. The actual size of the training set is unclear. I assume you used augmentation to produce 3600 samples from 100 images. Which augmentation steps were involved? This is never discussed.
-Line 166 mentions the use of a least squares approach, but these are 2D maps you are describing, so some type of gradient descent must have been involved given the large parameter space. Can you explain further?
-Line 177 mistakenly references Figure 2; I believe this sentence should be citing Figure 5.
-I suggest listing the segmentation hyperparameters on line 176 and listing the referenced modifications on Line 205.
-The terms in Eqs. 3-5 are never defined. What are c, d, scale, and swift?
-Figure 8 should follow, not precede, your explanation of the seven tree masking steps. Otherwise, the images lack context.
-Line 349 references collecting a continuous sequence of frames. However, on line 339, you suggest the user only needs to photograph each tree once, which seems like a contradiction. Can you elaborate?
-Figure 18 is a little confusing, since it is described as the ‘test set’, and I suggest moving it to the appendix. Why does it include non-tree images and from where were they acquired? VOC 2007 is briefly mentioned but its purpose remains a little unclear.
-Is there any overlap between Tables 4, 5, and 6? Are these 20 random samples? Are they the same 20 samples (taken from 110, I assume) in each case?
-The Haglof Sweden 447 Vertex IV device should be mentioned earlier and explained in more detail.
-I didn’t understand your description of ‘breast’ height models in Section 4.3. Is this referring to the height of the camera, the height of the trees, or something else?
-Is there a reason Figure 22 shows the maximum and average error but not the minimum?
-The acknowledgements section is included twice by mistake.
I believe the paper will be ready for publication once these issues are addressed. Well done.
The linguistic quality of the text is high. There are a few typographical errors (identified above), but I don't have any major concerns regarding the writing style or English content. The paper was very well written.
